# Triacylglycerides and Cholesterol in Organic Milk from Chiapas, Mexico

**DOI:** 10.3390/ani12101292

**Published:** 2022-05-18

**Authors:** José Jesús Pérez González, Marcela Guillermina Ortiz Romero, Beatriz Schettino Bermúdez, Jorge Luis Ruíz Rojas, Claudia Cecilia Radilla Vázquez, Nelly Molina Frechero, Rey Gutiérrez Tolentino

**Affiliations:** 1Departamento de Producción Agrícola y Animal, Universidad Autónoma Metropolitana Unidad Xochimilco, Mexico City 04960, Mexico; jjperez@correo.xoc.uam.mx (J.J.P.G.); schettinob@yahoo.com.mx (B.S.B.); cradilla@correo.xoc.uam.mx (C.C.R.V.); nmolinaf@hotmail.com (N.M.F.); 2División de Ciencias Biológicas y de la Salud, Universidad Autónoma Metropolitana Unidad Xochimilco, Mexico City 04960, Mexico; marshall_gor@hotmail.com; 3Facultad de Medicina Veterinaria y Zootecnia, Universidad Autónoma de Chiapas, Tuxtla Gutiérrez 29060, Mexico; jlrojas89@hotmail.com

**Keywords:** triacylglycerides, cholesterol, organic milk, Mexico

## Abstract

**Simple Summary:**

The profile of triacylglycerides and sterols in fat of raw cow’s milk produced under organic conditions in Tecpatán, Chiapas, Mexico was determined. The triacylglyceride values described a bimodal behavior, with maximums at C38 in the first mode and, between C50 and C52, the second mode. Cholesterol presented the highest percentage (96.41%) of sterols.

**Abstract:**

The study of milk fat composition is a priority topic at the international level; however, there are few studies on the composition of triacylglycerides (TAG) and sterols in cow’s milk produced in organic production systems. The objective of this study was to determine the profile of TAG, cholesterol, and other sterols in the fat of raw cow’s milk produced under organic conditions in the municipality of Tecpatán, Chiapas. Every month for one year, milk samples were obtained from three production units (PU 1, 2 and 3) and from the collecting tank (CT) of the municipality (12 months × 4 = 48 samples), in accordance with Mexican regulations. Milk fat was extracted by detergent solution and TAG and sterol analyses were performed by gas chromatography with a flame ionization detector and capillary columns. Chromatographic analyses identified and quantified 15 TAG in all milk fats, from C26 to C54, with a bimodal behavior; the maximum value (% *w*/*w*) for the first mode was located at C38 (14.48) and, for the second mode, C50 and C52 were considered with values of 11.55 and 11.60, respectively. Analysis of variance (ANOVA) followed by Tukey’s test only yielded significance (*p* < 0.05) for C26; most TAG values over time showed homogeneous variability. Cholesterol, brassicasterol, and campesterol were also determined; ANOVA did not show statistical significance (*p* ≥ 0.05) between them in the production units and collecting tank. Cholesterol had the highest percentage of the sterols with a mean value of 96.41%. The TAG and cholesterol profiles found in this study were similar to those reported in other countries.

## 1. Introduction

The production of organic milk is a worldwide reality that is constantly growing; of the total bovine milk produced, it is estimated that 1.9% is obtained from organic production systems, that is to say, around 16 billion liters. In Mexico, during 2019, 12,300 billion liters of cow’s milk were produced, of which it is estimated that 22 million liters were organic milk, equivalent to 0.18% participation [1,2]. However, even though this type of product is gaining ground, there are still gaps to be filled in terms of quality assessment. Studies have been carried out to describe the compositional and microbiological aspects, the presence of residues and contaminants and, to a lesser extent, of functional substances such as the study by Kourimska et al. [3], where 528 milk samples of Holstein cows from organic farms in the Czech Republic were analyzed; the results demonstrated contents of fat (4.03 g 100 g^−1^), protein (3.28 g 100 g^−1^), lactose (4.80 g 100 g^−1^), non-fat solids (8.66 g 100 g^−1^), caseins (2.61 g 100 g^−1^), and 218,000 somatic cell count, which were very similar to those obtained for conventional milk. Regarding the presence of waste and pollutants, Welsh et al. [4] divided the territorial space of the United States into nine regions. They evaluated the presence of organochlorine and organophosphorus pesticides and microbial inhibitors in 34 organic milk samples distributed in the nine regions; the results indicated the presence of three organochlorine pesticides in most of the samples, hexachlorobenzene (100%), ppDDE (100%), ppDDT (94%); they found no residues of organophosphorus pesticides and of penicillins, tetracyclines, sulfonamides, ionophores, and pyrimidine. In another study [5] conducted in the Canary Islands, in organic milk of ten commercial brands, it was also found that hexachlorobenzene was present in most of the samples analyzed; in addition, polychlorinated biphenyls similar to dioxins were found in considerable quantities, including above 25 pg g^−1^ fat base, a value well above the value established by the World Health Organization for concentration of toxic equivalents. These data show that the organic milk tested is not free of toxic substances, although it showed lower concentrations of residues and contaminants than conventional milk. Regarding the content of functional substances, such as bioactive lipids, scientific studies have recognized the benefits of the consumption of certain fatty acids; butyric acid (C4) has been recognized for its antitumor effect on prostate, breast, and colon; caproic (C6), caprylic (C8), and capric (C10) acids have been associated with the inhibition of microbial and viral growth and dissolution of cholesterol deposits in in vitro tests and in experimental animals. Conjugated linoleic acid (CLA) has been reported to have anticancer and antiteratogenic properties and has even been associated with weight loss, although the data obtained come from studies in cell cultures and experimental animals [6,7]. In a study of organic milk from ten commercial brands offered in the north of England, the contents of short-, medium-, and long-chain, saturated and unsaturated fatty acids were recorded. The contents of conjugated linoleic, eicosapentaenoic and docosahexaenoic acids were considerably high—7.9, 0.8, and 0.8 (g kg^−1^ of total fatty acids), respectively [8].

The aforementioned research shows that there are published studies on the nutritional content, microbiological quality, and content of pernicious and functional substances in organic cow’s milk, but worldwide studies on the composition of triacylglycerides and cholesterol content are scarce, and, in Mexico, no research has been published describing the annual trend of the triacylglyceride and cholesterol profile in organic milk produced in Chiapas. Therefore, and due to the relevance of the subject at the national and international level, the purpose of this study was to characterize the profile of triacylglycerides, cholesterol, and other sterols in organic milk produced in Tecpatán, Chiapas, Mexico, by capillary gas chromatography during a year.

## 2. Materials and Methods

### 2.1. Sample Source

The milk samples were obtained from Creole breed cattle (Zebu-Swiss Brown) from the municipality of Tecpatán, Chiapas. This municipality has about 3500 cows between three and seven years of age, distributed among 80 ranches that meet the characteristics of organic production units (PU). Cattle are grazed mainly on Insurgente (*Brachiaria brizantha*), Mombaza (*Panicum maximum*), Rajador *(Lysiloma divaricatum*), Cabezón (*Panicum virgatum* L.), and Mulato (*Bracharia hybrid* 36087) grasses.

Tecpatán is located between parallels 16°59′ and 17°24′ north latitude; meridians 93°14′ and 93°13′ west longitude; and altitude between 0 and 1200 m. It is bordered on the north by the states of Veracruz de Ignacio de la Llave and Tabasco, the interstate CH-T area and the municipalities of Ostuacán and Francisco León; on the east by the municipalities of Francisco León and Copainalá; on the south by the municipalities of Copainalá, Berriozabal and Ocozocuautla de Espinoza; and on the west by the municipalities of Ocozocuautla and Cintalapa and the states of Oaxaca and Veracruz de Ignacio de la Llave. It occupies 1.68% of the state’s surface and has 37,543 inhabitants. The climate is warm and humid with abundant rainfall in the summer (50.37%), warm and humid with year-round rainfall (49.11%), and semi-warm and humid with year-round rainfall (0.52%) [9].

### 2.2. Sample Collection

Milk samples were obtained for one year, at 30-day intervals. One liter of milk was collected immediately after milking from three PU out of 80 located in Tecpatán. In addition, one liter from the collecting tank (CT) and cooling tank available to all organic milk producers in Tecpatán, and a total of four liters per month for one year (48 samples in total). The samples were taken to the laboratory for analysis under refrigerated conditions and in accordance with Mexican regulations [10].

### 2.3. Milk Fat Extraction

The extraction of milk fat was carried out by the technique described by Frank et al. [11]; in a 500 mL volumetric flask, 250 mL of milk were mixed with 250 mL of detergent solution, and 50 g of sodium hexametaphosphate (Meyer, Mexico) plus 24 mL of Triton X100 (Sigma-Aldrich, St. Louis, MO, USA) in one liter of distilled water. The flask was placed in a water bath at 90 °C ± 3 °C and every 15 min for three times was shaken by inversion, until the fat was separated. The fat was extracted with a Pasteur pipette and filtered in the presence of sodium sulfate (J. T. Baker, State of Mexico, Mexico); it was kept frozen (−4 °C) until analysis.

### 2.4. Analysis of Triacylglycerides by Gas Chromatography with a Flame Ionization Detector

Milk fat consists of 95–98% triacylglycerides (TAG) [12]. The determination of triacylglycerides was performed according to Firestone [13]; 50 mg of the anhydrous milk fat were weighed into 5 mL amber vials with tightly sealed caps and dissolved in 2.5 mL of hexane (J. T. Baker, Phillipsburg, NJ, USA); 1.2 µL of the supernatant was taken and injected, in duplicate, into the gas chromatograph.

### 2.5. Chromatographic Conditions

A Shimadzu GC 2010 Plus gas chromatograph (Tokyo, Japan) with a 15 m long Restek capillary column with 0.53 mm internal diameter and 0.25 µm layer thickness (Rtx-50, 50% phenyl-50% methyl polysiloxane, Cat. No. 10522, Stockbridge, GA, USA) was used. Temperatures: 200, 340, and 360 °C of furnace, detector, and injector, respectively. Temperature program: T1 = 200 °C start (zero min), increased by 5 °C min^−1^ until reaching T2 = 300 °C, after 23 min increased by 3 °C min^−1^ until reaching T3 = 324 °C, maintained for 30 min, then increased by 5 °C min^−1^ until reaching T3 = 324 °C. The total running time was 81 min. Nitrogen was used as carrier gas at a pressure of 14 psi with a flow rate of 5.14 mL min^−1^; the injection was split type. The identification and quantification of the chromatographic signals (peaks) were carried out by the external standard method and using the Shimadzu GC Solution Crhomatography Data System Version 2.4 software. The standard used was a mixture of five triacylglycerides with 20% of each with purity greater than 99% of concentration 20 mg mL^−1^ in hexane of: tricaprylin (C24), tricaprin (C30), trilaurin (C36), trimyristin (C42) and tripalmitin (C48) (Lipid standard: Triglyceride mixture 100 mg, Supelco, No. Cat. 178-11, Bellefonte, PA, USA). The injection volume of the sample and standard was 1.2 µL.

The correction factors for TAG normalization and the equation of the line for the determination of the rest of TAG were obtained as described by Gutiérrez et al. [14], with the variant that the injection volume was 1.2 µL instead of 1 µL. The 100 mg ampule was dissolved in 5 mL of hexane. One microliter of the solution was injected 5 times to determine the retention time and the area percentage for each TAG. The average, minimum, and maximum retention times were calculated. The average area percentage was calculated and the correction factors were calculated considering the response factor for trilaurin (C36) as 1.0 and using the formula fx = CX/C36 × AC36/ACX, where fx = triacylglycerol × correction factor; CX = standard triacylglycerol × (mg mL^−1^) concentration; C36 = trilaurin concentration (mg mL^−1^); AC36 = trilaurin area; and ACX = standard triacylglycerol × area. The correction factors should not be greater than 1.01.

### 2.6. Cholesterol Analysis by Gas Chromatography with a Flame Ionization Detector

Cholesterol analysis was carried out on the unsaponifiable fraction of milk fat, following the official method AOAC 976.26-1977 [15], which also allows the extraction of other sterols. In addition, 2.5 g of fat were weighed and saponified for 25 min with 25 mL of 5% (*w*/*w*) methanolic potash; after cooling, the fat was transferred to a separatory flask and 30 mL of petroleum ether (J.T. Baker, Phillipsburg, NJ, USA) were added, the mixture was shaken by inversion for 5 min, then washed with distilled water, and the ether containing cholesterol and other sterols was extracted. The washing was repeated twice to ensure the best extraction. The ethereal extract was filtered through anhydrous sodium sulfate to remove the possible presence of water and brought to dryness in a rotary evaporator. Finally, the dried sample was reconstituted with 3 mL of hexane for injection into the gas chromatograph.

### 2.7. Chromatographic Conditions

A gas chromatograph with flame ionization detector GC-2010 Plus (Shimadzu, Kyoto, Japan) was used with a Restex capillary column of 30 m length, 0.25 mm internal diameter and 0.25 µm layer thickness (Rtx Crossbond, 5% diphenyl-95% dimethyl polysiloxane, Cat. No. 10223, Stockbridge, GA, USA). Temperatures: 120, 320, and 320 °C of furnace, detector, and injector, respectively. Temperature program: T1 = 120 °C for one min, then increased 35 °C min^−1^ to T2 = 190 °C, then increased 15 °C min^−1^ to T3 = 305 °C, held for 29 min, for a total of 39 min run. Nitrogen was used as carrier gas at a pressure of 9 psi with a flow rate of 9 mL min^−1^. The injection type was splitless. The identification and quantification of the chromatographic signals were carried out by the external standard method and using the Shimadzu GC Solution Crhomatography Data System Version 2.4 software. The standard used was a mixture of sterols: cholesterol (cholesterol, 5-cholesten-3β-ol, Sigma, No. Cat. C-8667, USA), brassicasterol (5,22-cholestadien-24β-methyl-3β-ol, Sigma, No. Cat. B4836, USA), campesterol (24α-methyl-5-cholesten-3β-ol from soybean 65% approx., Sigma, No. Cat. C-5157, USA), stigmasterol (3β-hydroxy-24-ethyl-5,22-cholestadiene, Sigma, Cat. No. 47132, USA), and beta sitosterol (24β-ethyl cholesterol from soybeans, Sigma, Cat. No. 9889, USA). Injection volumes were 1 and 0.8 µL for standard and samples, respectively.

### 2.8. Statistical Analysis

The variables measured were triacylglycerides and cholesterol, among other sterols.

The study was descriptive and longitudinal for one year, with monthly observation. The data obtained were used to create a database that was subjected to exploratory analysis to observe the distribution behavior and, if applicable, outliers. The TAG profile was determined by simple linear regression, selecting as a dependent variable the retention time of each TAG signal and as an independent variable the corresponding number of carbons. Likewise, grouping, and inferential statistical tests were applied to contrast mean values and find similarities between the contents of TAG, cholesterol, and other sterols in the analyzed milks from the three production units (PU) and collecting tank (CT). All analyses were performed using the statistical software IBM^®^ SPSS^®^ version 24.0 for Windows (Armonk, NY, USA) [16].

## 3. Results and Discussion

### 3.1. Triacylglycerides

The analysis of triacylglycerides (TAG) by capillary gas chromatography defined a typical graphical profile of the standard mixture of the five TAGs, tricaprylin, C24; tricaprin, C30; trilaurin, C36; trimyristin, C42 and tripalmitin, C48. Likewise, the TAG present in the milk fat studied was defined graphically, going through the spectrum of low, intermediate, and high molecular weights, from C24 to C54 (Figure 1). By visual inspection, small chromatographic signals (peaks) are observed in C26, C28, C30, C32, C44, and C54. This chromatographic behavior is in accordance with that reported by other researchers in studies on bovine milk from Germany, Spain, Mexico, and India [14,17,18,19].

The mean values, standard deviations, and confidence intervals for the means of TAG C26 to C54 present in the milks of PU 1, 2, 3, and CT are shown in Table 1. One-way analysis of variance followed by Tukey’s 95% test yielded statistical significance only at C26 (*p* < 0.05), PU1 and CT were different, but PU 2 and 3 were statistically similar to each other and to PU1 and CT. The variability of the data is very small, the maximum standard deviation (SD) was obtained in C52 of PU1 with a value of 2.42, but the overall SD in C52 was 1.87. In a study conducted on cow’s milk produced in conventional systems in different regions of Mexico [14], C52 was also found to have a higher variability (SD = 5.29), much higher than in this study, probably due to the larger distance between production zones with the characteristics of each one of them. In another work carried out in Dutch milk fat produced in an organic system [20], less variability was reported in most of the 16 TAG, with a minimum SD for C26 equal to 0.02 and a maximum in C52 and 54 equal to 0.56, while, in this study, the minimum SD was obtained in C30 of PU2 with a value of 0.09. The overall means (% *w*/*w*) and SD values of TAG of the milk analyzed in this work together with other data reported in the literature are described in Table 2 [14,19,20,21]. Production conditions are particular to each of the regions (breed, age, climate, facilities, production system, feeding), so there is some variation in TAG content, although the profile is similar.

Advances in the analysis of TAG present in milk fat contribute to the knowledge of the composition and distribution of fatty acids (FA) in glycerol. More and more efforts are being made to understand the stereospecific composition of FA in the TAG glycerol molecule, due to the interest aroused by the discovery of functional properties of some fatty acids. Short- and medium-chain FA such as butyric (C4), caproic (C6), caprylic (C8), and capric (C10) are recognized to have positive effects on human health, not only as anticarcinogens, but also as antibacterial and antiviral agents. Short-chain FA are easily absorbed in the intestine and passed into the circulatory system without TAG resynthesis; they are used as a source of quick energy. There are also known benefits of unsaturated long-chain FA such as C18:1 11t, C18:2 9c-12c, C18:3 9c-12c, and conjugated linoleic acid (CLA) and its isomers in atherosclerosis, cardiovascular disease, cancer, diabetes, and hypertension [7].

Another aspect other than health, but no less important, is the use of the TAG profile in the detection of foreign fat in milk fat, i.e., it makes it possible to authenticate milk and milk products. There have been published works that detect milk adulterations with TAG ratios and multiple linear regression and linear discriminant equations, to 2% addition of non-dairy fat in milk and milk products [14,22]. As a result, organic dairy products, being more expensive than conventional ones, are very attractive for the illegal practice of adulteration.

The distribution of TAG contents in the milks of PU 1, 2, 3, and CT sketches a bimodal behavior, with a maximum value at C38 for the first mode and, since the values are so close at the peak of the second mode, the peak occurs at C50 and C52 (Figure 2). The minimum value between both modes was located at C44. As can be seen, the lines of the graph are very close together due to the low dispersion of the data. Other authors have reported this bimodal behavior in bovine milk fat produced in organic and conventional systems [20,22]; however, it is not the same in milk from other species such as buffalo, donkey, goat, human, and sheep. The maximum values (% *w*/*w*) of the two modes in buffalo milk are found in C38 (15.52) and C50 (9.96); in donkey C44 (10.15) and C52 (16.22) and sheep C40 (13.62) and C52 (8. 46), but the TAG profile in goat and human milks do not reflect bimodal behavior; rather, a single mode is observed, with a maximum in C52 in both cases, with values of 15.23 and 32.71 (% *w*/*w*), respectively [19,23]. PU 1, 2, 3, and CT samples showed a similar profile of TAG groups where C36, C38, and C40 were the major TAGs, in agreement with the reported data by Fontecha et al. [24] in different milk fats.

The analysis of variance did not show statistical significance in TAG contents between PU 1, 2, 3, and CT, except for C26 (*p* < 0.5), which is probably due to the breed of the cows, since it is similar in all production units, and to the presence of rain all year round, which guarantees homogeneous pasture for the animals. In January, October, and November, PU3 showed higher values than the other PU and CT, and, in July and August, it showed lower values together with PU1. In general, the trend was constant, except for the variations mentioned above (Figure 3). This behavior has been documented in several studies; in one of them, TAG was analyzed in winter and summer in pasteurized organic milk fat from the Netherlands, identifying and quantifying from C24 to C54; the results left show that all contents (% *w*/*w*) of TAG were similar, only distant values without reaching statistical significance were found in C40, 9.78 winter vs. 10.11 summer, and C54, 3.00 winter vs. 3.77 summer [22]. In another annual study in organic milk from Denmark, some trend of variation over time was observed, contents (% *w*/*w*) were recorded lower in C24 to C38 and C44 to C48 from May to August and higher in 50 to 54 compared to those obtained from September to December [21]. Seasonal differences in composition and contents have also been identified; Capuano et al. [25] studied the concentration of TAG in Danish organic milk in different seasons of the year; their results detected statistical difference in spring winter, mainly due to feeding strategies, including an increase in C40, C50, and C52 when cows grazed longer.

Table 3 shows the Pearson correlations between TAG, with significance at 95 and 99%. A positive correlation was observed for TAG C30 to C36, except for C26 with C36, which was negative. From C36 to C54, the correlations were negative and positive. These results coincide, for the most part (excluding C26), with those found in Dutch milks, where positive correlations were obtained for TAG groups C26–C30 (r > 0.80), C32–C36 (r > 0.56), C42–C46 (r > 0. 61), and C50–C54 (r > 0.73), and negative correlations between some TAG of the groups C50–C54 with some of the groups C32–C36 (r > 0.56), C42–C46 (r > 0.44); as well as between C50 with some TAG of the group C26–C30 (r > 0.44) [25].

### 3.2. Cholesterol and Other Sterols

The sterol standard was composed of cholesterol, brassicasterol, campesterol, stigmasterol, and beta sitosterol. Since no statistical significance (*p* ≥ 0.05) was observed in the milk from PU 1, 2, 3, and CT, the overall values obtained in the chromatographic analyses are reported, which allowed the identification and quantification of only three of the five sterols, cholesterol, brassicasterol, and campesterol with 96.41, 2.0, and 1.59 (% for each of the total sterol content), respectively (Figure 4A). The literature reports that cholesterol is the major sterol in cow’s milk, in amounts ranging between 20 and 30 mg L^−1^ [12], representing about 97% of the total sterols; the rest are trace amounts of sterols such as lanosterol and some characteristic of vegetable fats such as brassicasterol, campesterol, stigmasterol, and beta sitosterol. Karrar et al. [26] studied the composition of sterols in milk fat from different species. Cholesterol was the major sterol in milk fat of all studied species. The cholesterol content of camel, goat, cow, donkey, and yak milk fat was 2.11, 2.08, 2.03, 1.87, and 1.68 mg g^−1^ fat, respectively. After cholesterol, desmosterol was quantitatively the main sterol in camel and donkey milk fat, 0.25 and 0.11 mg g^−1^, respectively, but it was not detected in cow, goat, and yak milk fat. Beta sitosterol was detected only in donkey milk fat. The discrepancies in sterols composition between these studies might be related to variations in the conditions of the animal and production systems, such as age, breed, feeding, season, and environment. In general terms, it is accepted that the level of serum cholesterol is a risk factor for coronary heart disease, so it is recommended to reduce the consumption of fat and in particular saturated fat and cholesterol; although there are relatively recent studies that have not found any relationship between milk fat consumption and cardiovascular disease [27]. On the other hand, milk fat plays an important role in economic aspects because it is more expensive than other fats (butter vs. margarine) and because of the content of bioactive lipids such as butyric and conjugated linoleic acid, to which anticarcinogenic effects are attributed, making it very attractive for the food industry. However, it has been documented that adulteration of dairy products is a reality [14], i.e., a dairy product may have undergone the substitution of its original fat by another cheaper fat such as vegetable oils, partially hydrogenated, and if this action is not declared on the product label, it is considered an adulteration. In a recent study, it was found that 50% of pasteurized and ultra-pasteurized commercial milk samples (*n* = 17) in Kazakhstan were adulterated in their fat fraction. In this study, the sterol profile was determined by gas chromatography with mass-spectrometric detection [28]. It is here where the sterol profile takes on great relevance, since it facilitates the detection of vegetable fat in milk and dairy products, and when higher percentages of vegetable sterols (>5%) are found, adulteration is demonstrated. Of course, this practice in organic milk is more attractive since the cost of organic milk and dairy products is at least 50% higher than the cost of conventional products.

The distribution of sterols (% of total sterols) over time was constant, as was the TAG content, and no seasonal variation could be fully identified. The lowest values for cholesterol were obtained in August, 95.6, October, 95.7, and November, 95.6%, and the highest in March, 97.4, June, 97.0, and September, 97.0%. The percentages for brassicasterol and campesterol ranged between 1.14–2.77 and 1.38–2.07, with means 2.0 and 1.59, respectively (Figure 4).

## 4. Conclusions

In this work, triacylglyceride and sterol profiles were characterized in samples of milk produced with organic characteristics in the state of Chiapas. Fifteen TAG of low, medium, and high molecular weights, from C26 to C54, were identified and quantified. Cholesterol was the sterol with the highest abundance (96.41%), followed by brassicasterol (2.0%) and campesterol (1.69%). Statistical analysis yielded no significance (*p* ≥ 0.05) for TAG and sterols, except for C26 (*p* < 0.05). Pearson’s correlation analysis reflected positive correlations (at 95 and 99%) in the TAG group C30 to C36, except for C26 with C36, which was negative. Statistical regularity over time was observed in the TAG and sterol contents in the milk analyzed.

## Figures and Tables

**Figure 1 animals-12-01292-f001:**
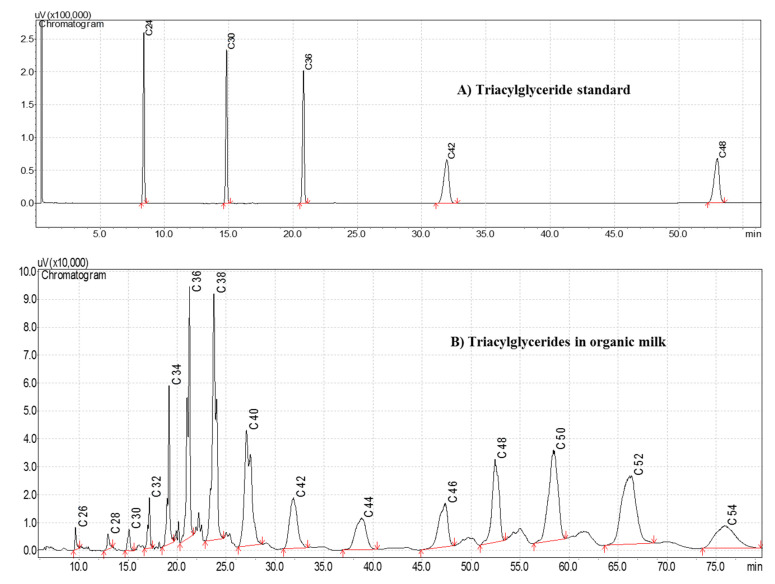
Chromatograms of triacylglycerides, injection volume 1.2 µL. (**A**) chromatogram of the triacylglyceride standard; (**B**) chromatogram of triacylglycerides from organic milk fat produced in Tecpatán, Chiapas.

**Figure 2 animals-12-01292-f002:**
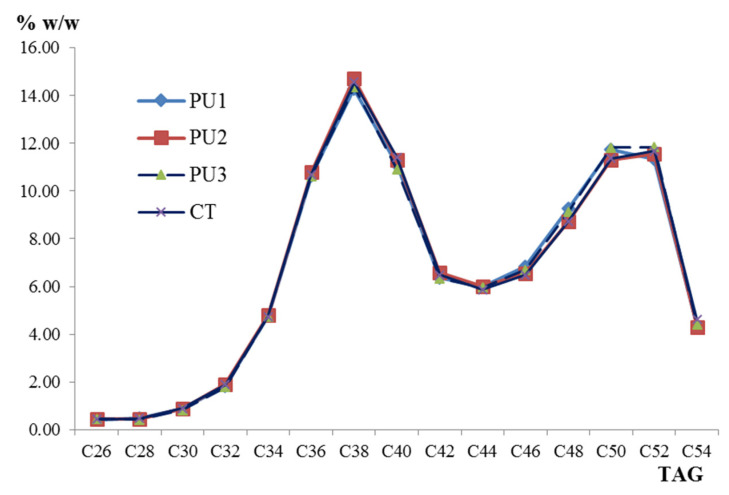
Distribution of mean values of triacylglycerides (TAG, % *w*/*w*) in organic bovine milk from Tecpatán, Chiapas (PU 1, 2, 3, and CT).

**Figure 3 animals-12-01292-f003:**
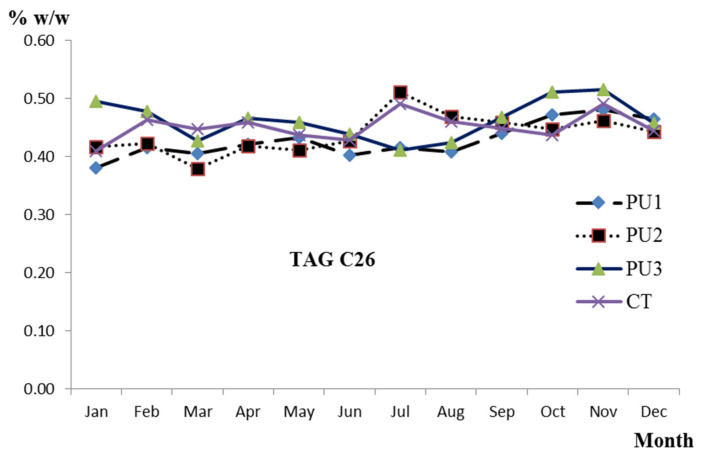
Distribution trend over time of mean values (% *w*/*w*) of triacylglyceride C26 in organic milk produced in Tecpatán, Chiapas (PU 1, 2, 3, and CT).

**Figure 4 animals-12-01292-f004:**
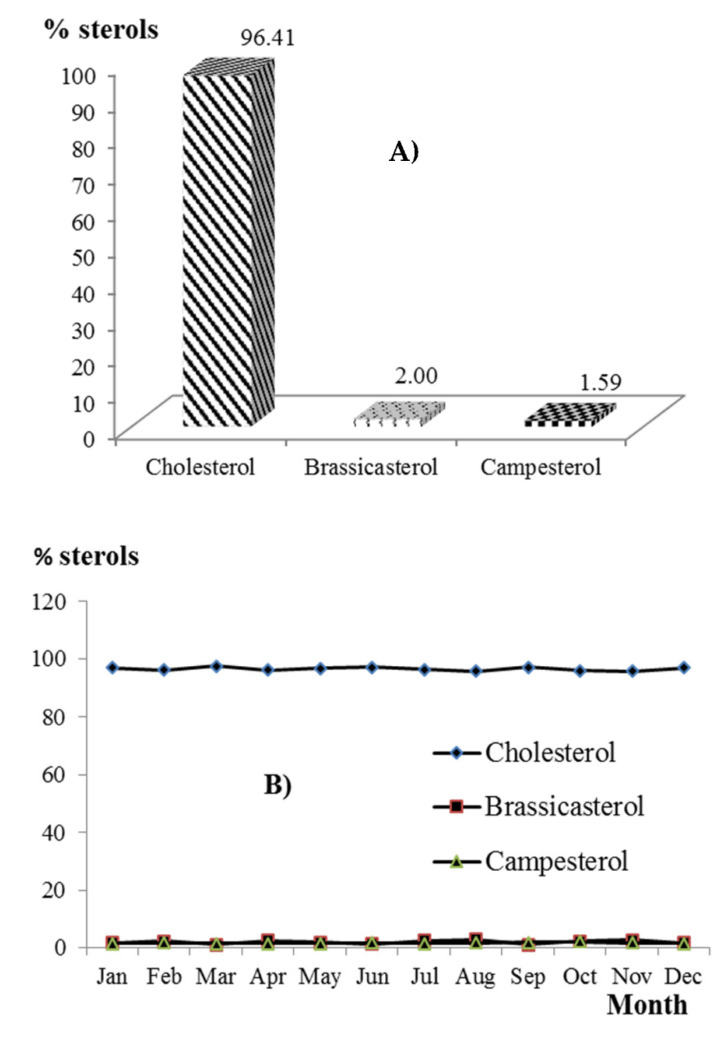
(**A**) Mean values (% of total sterols) of cholesterol, brassicasterol, and campesterol in milk produced in Tecpatán, Chiapas; (**B**) distribution of sterols over time.

**Table 1 animals-12-01292-t001:** Composition of triacylglycerides (% *w*/*w*) in organic milk from the municipality of Tecpatán, Chiapas.

TAG	PU1		PU2		PU3		Collector Tank	
	^§^ ±SD	CI	±SD	CI	±SD	CI	±SD	CI
C26	0.43 ^a^/0.29	(0.41,0.45)	0.44 ^ab^/0.34	(0.42,0.46)	0.46 ^ab^/0.35	(0.44,0.49)	0.45 ^b^/0.23	(0.44,0.47)
C28	0.49 ^a^/0.96	(0.43,0.55)	0.46 ^a^/0.32	(0.44,0.48)	0.43 ^a^/0.78	(0.38.0.48)	0.47 ^a^/0.54	(0.44,0.51)
C30	0.87 ^a^/0.15	(0.77,0.97)	0.88 ^a^/0.09	(0.83,0.94)	0.82 ^a^/0.16	(0.72,0.92)	0.90 ^a^/0.13	(0.82,0.98)
C32	1.83 ^a^/0.33	(1.52,2.03)	1.89 ^a^/0.17	(1.78,2.00)	1.79 ^a^/0.32	(1.59,1.99)	1.89 ^a^/0.28	(1.71,2.06)
C34	4.77 ^a^/0.87	(4.22,5.33)	4.81 ^a^/0.44	(4.53,5.09)	4.74 ^a^/0.65	(4.31,5.15)	4.74 ^a^/0.57	(4.38,5.10)
C36	10.66 ^a^/1.18	(9.90,11.41)	10.79 ^a^/0.71	(10.34,11.24)	10.60 ^a^/0.81	(10.08,11.11)	10.69 ^a^/0.84	(10.15,11.22)
C38	14.27 ^a^/0.74	(13.81,14.74)	14.72 ^a^/0.58	(14.34,15.09)	14.36 ^a^/0.37	(14.12,14.59)	14.56 ^a^/0.30	(14.37,14.75)
C40	11.08 ^a^/1.03	(10.42,11.74)	11.31 ^a^/0.31	(11.11,11.51)	10.92 ^a^/0.48	(10.61.11.22)	11.43 ^a^/0.51	(11.10,11.75)
C42	6.37 ^a^/0.58	(6.0,6.7)	6.59 ^a^/0.36	(6.36,6.82)	6.33 ^a^/0.53	(5.99,6.67)	6.49 ^a^/0.51	(6.17,6.81)
C44	6.01 ^a^/0.92	(5.42,6.59)	6.02 ^a^/0.45	(5.74,6.31)	5.95 ^a^/0.67	(5.52,6.37)	5.88 ^a^/0.72	(5.42,6.33)
C46	6.83 ^a^/1.01	(6.19,7.48)	6.55 ^a^/0.43	(6.28,6.82)	6.70 ^a^/0.55	(6.35,7.05)	6.48 ^a^/0.67	(6.05,6.90)
C48	9.29 ^a^/0.98	(8.66,9.91)	8.72 ^a^/0.34	(8.50,8.94)	9.12 ^a^/0.54	(8.78,9.47)	8.72 ^a^/0.52	(8.38,9.04)
C50	11.73 ^a^/0.71	(11.27,12.17)	11.31 ^a^/0.39	(11.06,11.56)	11.82 ^a^/0.98	(11.19,12.44)	11.36 ^a^/0.52	(11.03,11.70)
C52	11.35 ^a^/2.42	(9.81,12.88)	11.54 ^a^/1.31	(10.71,12.38)	11.83 ^a^/1.89	(10.63,13.03)	11.68 ^a^/1.88	(10.48,12.87)
C54	4.34 ^a^/1.71	(3.26,5.43)	4.29 ^a^/0.84	(3.76,4.83)	4.42 ^a^/1.15	(3.69,5.15)	4.61 ^a^/1.37	(3.74,5.48)

TAG: triacylglyceride, PU: production unit; ^§^: arithmetic mean, SD: standard deviation, CI: confidence interval for the arithmetic mean at 95%. Different letters in means of the same row indicate 95% difference.

**Table 2 animals-12-01292-t002:** Means (% *w*/*w*) and standard deviations of triacylglycerides in organic and conventional bovine milk fat from different countries.

TAG	This Study	Denmark ^a^	The Netherlands ^b^	India ^c^	Mexico ^d^
C24 to C30	-	0.30 ± 0.16	-	-	-
C26	0.45 ± 0.03	-	0.34 ± 0.02	0.26 ± 0.01	-
C28	0.46 ± 0.07	-	0.78 ± 0.04	0.56 ± 0.01	0.45 ± 0.19
C30	0.87 ± 0.13	-	0.43 ± 0.10	0.95 ± 0.01	0.58 ± 0.51
C32	1.85 ± 0.28	2.60 ± 0.22	2.86 ± 0.15	1.87 ± 0.03	1.45 ± 0.52
C34	4.77 ± 0.63	5.80 ± 0.25	6.13 ± 0.25	4.49 ± 0.03	3.41 ± 0.97
C36	10.68 ± 0.88	10.50 ± 0.58	10.91 ± 0.29	9.29 ± 0.02	7.11 ± 1.57
C38	14.48 ± 0.54	12.00 ± 0.34	12.27 ± 0.14	12.62 ± 0.04	11.26 ± 1.98
C40	11.18 ± 0.65	9.30 ± 0.35	9.91 ± 0.17	10.64 ± 0.04	10.68 ± 3.24
C42	6.44 ± 0.50	6.50 ± 0.58	7.11 ± 0.17	6.14 ± 0.07	7.26 ± 3.26
C44	5.96 ± 0.69	6.70 ± 0.53	6.68 ± 0.24	5.34 ± 0.07	5.27 ± 1.67
C46	6.64 ± 0.69	7.50 ± 0.51	7.34 ± 0.23	6.29 ± 0.05	6.59 ± 1.38
C48	8.96 ± 0.67	9.2 ± 0.34	8.79 ± 0.20	8.59 ± 0.03	9.34 ± 1.53
C50	11.55 ± 0.70	11.60 ± 0.56	10.93 ± 0.31	12.17 ± 0.07	13.43 ± 3.27
C52	11.60 ± 1.86	11.10 ± 1.27	9.44 ± 0.56	12.87 ± 0.11	14.64 ± 4.98
C54	4.41 ± 1.27	4.10 ± 0.85	4.67 ± 1.56	7.65 ± 0.07	8.61 ± 3.52

^a,b,c^: organic bovine milk; ^d^: conventional bovine milk.

**Table 3 animals-12-01292-t003:** Correlations between TAG of milk (PU 1, 2, 3, and CT) produced in Tecpatán, Chiapas.

TAG	C26	C28	C30	C32	C34	C36	C38	C40	C42	C44	C46	C48	C50	C52	C54
C26	1														
C28	−0.3 *	1													
C30	−0.3 *	0.8 **	1												
C32	−0.2	0.8 **	0.9 **	1											
C34	−0.1	0.8 **	0.9 **	0.9 **	1										
C36	−0.2	0.6 **	0.8 **	0.9 **	0.9 **	1									
C38	−0.5 **	0.1	0.3 *	0.2	0.2	0.4 **	1								
C40	−0.3 *	−0.1	−0.1	−0.2	−0.5**	−0.4**	0.5**	1							
C42	−0.1	0.7 **	0.9 **	0.9 **	0.9 **	0.8 **	0.2	−0.3 *	1						
C44	0.02	0.6 **	0.8 **	0.8 **	0.9 **	0.8 **	−0.05	−0.6 **	0.9 **	1					
C46	0.2	0.5 **	0.5 **	0.7 **	0.8 **	0.7 **	−0.3	−0.8 **	0.7 **	0.9 **	1				
C48	0.4**	0.1	0.1	0.2	0.4 *	0.3 *	−0.6 **	−0.9 **	0.3	0.6 **	0.8 **	1			
C50	0.4**	−0.7 **	−0.9 **	−0.8 **	−0.7 **	−0.7 **	−0.6 **	−0.2	−0.8 **	−0.5 **	−0.2	0.3 *	1		
C52	0.1	−0.7 **	−0.9 **	−0.9 **	−0.9 **	−0.9 **	−0.2	0.5 **	−0.9 **	−0.9 **	−0.9 **	−0.5 **	0.6 **	1	
C54	0.04	−0.6 **	−0.7 **	−0.8 **	−0.9 **	−0.9 **	−0.1	0.6 **	−0.8 **	−0.9 **	−0.9 **	−0.6 **	0.4 **	0.9 **	1

*: The correlation is significant at the 0.05 level (bilateral); **: The correlation is significant at the 0.01 level (bilateral).

## Data Availability

The data presented in this study are available on request from the corresponding author.

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
