# Peer review of "Triacylglycerides and Cholesterol in Organic Milk from Chiapas, Mexico"

_animals, 2022, doi:10.3390/ani12101292_

Round 1
Reviewer 1 Report
The scientific article Triacylglycerides and cholesterol in organic milk from Chiapas, 1 Mexico, provides a coherent text and addresses an important research topic.
The chapter on materials and methods, describes the working methodologies correctly, but needs improvement. However, the errors occurring in work do not constitute gross errors, but rather editorial errors.
Chapter Results and Discussion contains 4 figures and 3 tables, which clearly present the study results, statistics and chromatograms of triacylglycerides.
All reference works were titled in the body of the work.
Below is a list of the errors with commentary:
Line 45 - Please remove the spaces
Line 61-66 - Consider rephrasing this sentence
Line 82 - The course of may be redundant
Line 88-89 - Please give the full name, since it is used for the first time. Italics.
Line 98 - Please remove the spaces
Line 111, 114 etc. - For reagents and test apparatus enter in brackets (Company, city / state, country)
Line 112, 115 etc. - Suggests, not to put spaces before degrees celcius
Line 146 - How were the samples shaken, could you give any additional parameters.
Table 1 - For standard deviations it is suggested to use one symbol, either / or ± (a different notation appears in Tables 1 and 2 ).
Table 1 - "0.45b/0.23" Is the letter correct with this result? Please verify.
Please check References again and correct the errors.
Reviewer 2 Report
This is a correctly done research, only I don't really see the great importance of the set hypothesis, ie its purpose. It still brings some interesting information.
Reviewer 3 Report
This work characterizes the triacylglyceride and sterol profiles in milk samples produced in Chiapas state. The work begins with a correctly written theoretical introduction in which the authors correctly cite the scientific literature. The experimental part was well planned. The results of the research were properly discussed in the aspect of the available literature in the field of the studied problem. The conclusions are correct.
This positive evaluation of the manuscript does not mean that it cannot be improved. Namely, the authors should include information whether the GC-FD method used was validated? What was the quantification limit of individual triglycerides and cholesterol?
Reviewer 4 Report
The manuscript entitled "Triacylglycerides and cholesterol in organic milk from Chiapas, Mexico", emphasis on the profile of triacylglycerides and sterols in fat of raw cow's milk produced under organic conditions in Tecpatán, Chiapas, Mexico. The triacylglyceride values described a bimodal behavior, with maximums at C38 in the first mode and, between C50 and C52 the second mode. Cholesterol presented the highest percentage (96.41%) of sterols.
The manuscript is well written and holds some scientific merit. The research design is good, however, the author needs to revise the manuscript regarding English and Grammer corrections. There is a dire need to include recent work in this paper in terms of citation and references. The discussion part is weak and need improvement.
